# Validation of Health Education Material for Youth: A Step to Ensure Implementation Fidelity in Community-Based Interventions

**DOI:** 10.3390/healthcare8010008

**Published:** 2019-12-27

**Authors:** Shantanu Sharma, Faiyaz Akhtar, Rajesh Kumar Singh, Sunil Mehra

**Affiliations:** 1MAMTA Health Institute for Mother and Child, New Delhi 110048, India; akhtar@mamtahimc.org (F.A.); rksingh@mamtahimc.org (R.K.S.); dr_mehra@mamtahimc.org (S.M.); 2Independent Researcher, Department of Clinical Sciences, Lund University, Sweden

**Keywords:** health promotion, health communication, process assessment, teaching materials, validation studies, vulnerable populations

## Abstract

Health education materials such as flipbooks enhance learning and deliver key messages in a captivating mode. Validation of such materials is crucial to ensuring implementation fidelity. We conducted a study to achieve two objectives: (a) to develop two flipbooks, one each for adolescents and young married women (YMW); (b) to validate the flipbooks using five parameters, namely, content validity, construct validity, concurrent validity, relevance, and face validity. The study was a part of a community-based peer-led intervention on health, nutrition, and hygiene. The content validity and relevance were assessed by interviewing outreach workers (ORWs, *n* = 42) using self-administered five-point Likert scale-based tools. A pre- and post-intervention assessment of knowledge among adolescents (*n* = 100) and YMW (*n* = 50) across six out of 13 intervention sites was done to evaluate the construct validity. The two flipbooks contained 12 structured sessions with five key messages per session, in addition to illustrations, discussion points, and theme-based stories at the end of each session. The content and relevancy indices were ranked above 80% by ORW. There was a statistically significant increase in the knowledge scores of adolescents (*p* < 0.001) and YMW (*p* < 0.001) post intervention. The validation process helps in assessing the relevance and appropriateness of the education content for greater acceptance and responsiveness by the users.

## 1. Introduction

India lies on the cusp of demographic transition, with 27.5% of its population constituted by youths (15–29 years). At present, about 34% of India’s gross national income (GNI) is contributed by youths. Healthy, educated, skilled, and empowered youths have the potential to reap greater economic dividends for the nation [1]. Contrary to this need, the youths of the country, including adolescents and young people, are facing numerous health challenges such as early marriage, unwanted pregnancies, domestic or sexual violence, sexually transmitted infections, mental health problems, etc. Hence, there is a pressing need to implement effective youth-focused interventions to promote their positive development, along with the adoption of healthy behaviors, and to help them secure a healthy and productive future adult population [2]. Substantial evidence shows the effectiveness of health and nutrition interventions among adolescents and young married women (YMW), which have an alleged impact over generations [3,4]. Health education and promotion are effective modes of interventions for modifying health behavior [5]. The rate of literacy is around 86% among women (15–24 years) and 93% among men (15–24 years) in the country, which would make health education interventions more successful and sustainable [6].

Information, education, and communication (IEC) interventions, reflective of and responsive to the local culture and conditions, are an effective strategy for health education and promotion activities. IEC is a crucial step toward a social behavior change communication (SBCC) approach that motivates people to adopt and sustain healthy behaviors [7]. Printed IEC materials such as posters, flipbooks, and modules form the core of its strategies. They enhance learning, deliver key messages in a captivating mode, and serve as motivators, reminders for actions, and reinforcing tools for verbal communication [8]. Good-quality IEC materials are paramount for the effective translation of knowledge into actions. Newly developed education materials should be tested for validation to maximize their effectiveness [9]. The validation process should assess the adequacy of the material for its content, illustrations, indexing, learning, and cultural adequacy [10]. Most of the health programs develop or adopt IEC materials, but there are problems concerning the validation of their education content [11,12].

Validation of IEC materials is an essential step toward ensuring implementation fidelity, which is defined as the degree to which intervention is delivered as intended. The content of IEC materials is relevant for assuring adherence to an intervention, which is one of the five elements of implementation fidelity. Implementation fidelity is critical to the successful translation of evidence-based theories into actions [13,14,15]. Flipbooks are an education material carrying a series of ideas sequentially to help disseminate information in an interesting and engaging manner [16]. Many such IEC materials are available on the Government of India’s health and family welfare department’s website related to adolescence and YMW’s health. Still, issues such as adolescent mental health, life skills, marital communication among YMW, women empowerment schemes, preconception care, and men’s health are inadequately covered in these materials [17,18]. We attempted to address issues of inappropriate validation forms, not so rigorous methodology, and validation by capable professionals in our flipbooks. Based on these assumptions, this study aimed to construct and validate flipbooks on health, nutrition, and hygiene information for adolescents and YMW.

## 2. Materials and Methods

The study was conducted using a stepwise approach to achieve two objectives: (1) development of flipbooks for adolescents and YMW on health, nutrition, and hygiene; (2) validation of the flipbooks using a framework. The study was a part of project “*JAGRITI*”, which was a community-based intervention spread across 13 districts of India. The objective of the project was to improve the knowledge, attitude, and practices of marginalized populations (adolescents, young married women, pregnant and lactating women) on health, nutrition, and hygiene using a peer-led approach. The key approach of the intervention was health education sessions mediated by peer leaders using various IEC tools. The flipbooks were used as IEC tools to deliver health education sessions in the communities.

### 2.1. Development of Flipbooks

One of the three steps in the process of development of flipbooks was to obtain suggestions from subject experts for the conceptualization and effectiveness of flipbooks regarding its content, language, institution, layout, and captivation. The second step included a desk review, which was conducted by two public health specialists for the available literature and resources. Lastly, the needs and the key priority health and nutrition areas of interventions among adolescents and YMW were assessed by quantitative surveys and interviews, the details of which are beyond the scope of this article.

### 2.2. Validation of the Flipbooks

We developed the validation framework for the flipbooks considering all the aspects of content, construct, reliability, relevance, and concurrent validity (Table 1). Multiple rounds of discussions within the team and with the experts were made to make the IEC tools presentable, convenient, and succinct. For the evaluation of content validity and relevance, the sample size was calculated from the following formula:*n* = Zα^2^∙P(1 − P)/e^2^,(1)
where P is the expected proportion of participants indicating the adequacy of each item, and “e” represents the acceptable proportional difference compared to what would be expected. A confidence level of 95% was considered, and an assumption was made such that at least 50% of participants would rate this flipbook as appropriate. Thus, the values used for the calculation were Zα^2^ = 1.96, P = 0.70, and e = 0.11, and the sample size was 41. For construct validity (pre- and post-intervention assessment of project beneficiaries’ knowledge), the sample size was calculated using the following formula:*n* = 2 × [Z_crit_ √(2p3(1 − p3)) + Z_pwr_ √(p1(1 − p1) + p2(1 − p2))]^2^/D^2^(2)
where p1 and p2 are pre-study estimates of the two proportions to be compared from the previous study (p1, pre-intervention proportion of adolescents with satisfactory knowledge on health = 37%, and p2, post-intervention proportion of adolescents with satisfactory knowledge on health = 67%), D = p1 − p2, and p3 = (p1 + p2)/2, Z_crit_ = 1.960 and Z_pwr_ = 0.8 [19]. Thus, the required sample size was 45.

Hence, for the content validity and relevance assessment, 42 community outreach workers (ORW) were enrolled. Moreover, 100 adolescents (50 girls and 50 boys) and 50 YMW were enrolled for assessing the construct validity of the flipbooks using pre- and post-test analysis. In the sampling of the ORW and beneficiaries, a multi-stage random selection strategy was adopted. Six out of 13 intervention sites were selected randomly; furthermore, seven ORW and eight beneficiaries per group were selected randomly per site. A copy of the flipbook was handed over to all the participants for their understanding before participating in the assessment. Socio-demographic details of the participants were collected beforehand, including age, gender, educational status, type of community he/she belongs to, etc.

Face validity was assessed by obtaining suggestions from external reviewers. The reviewers included experts in gynecology, public health, and anthropology. All the data were collected from March to May 2018. Two of the five experts were interviewed for concurrent validity (“do the new flipbooks supplement the existing education material?”) of the flipbook. Verbal informed consent was obtained from the participants. The institutional ethical review board after a thorough review granted the approval for the project.

A self-administered five-point Likert scale was used for assessing content validity: strongly agree, agree, neutral, disagree, and strongly disagree. Moreover, the questionnaire contained open questions for suggestions and feedback. The 12 items in the scale included questions related to the relevance of the content, simplicity, and comprehensiveness of information, appropriateness of illustrations, duration of the sessions for the amount of text to be taught, etc. All the items were asked in the positive mode. Each item was considered adequate if the responses were “agree” or “strongly agree”. Two index approaches were used for assessing content validity. One index (I-1) was defined as the proportion of items agreed to be adequate by each participant. The other index (I-2) was defined as the proportion of participants agreeing to the adequacy of each item. Another self-administered five-point Likert scale was used to assess the relevance of the themes in the flipbooks. This 12-item scale questioned the relevance of each session in the context of the aims and objectives of the project.

The construct validity was assessed using a pretested, structured, and validated questionnaire for analyzing the increase in knowledge before and after the intervention. The intervention comprised 12 education sessions, which were conducted at the youth information centers in the area. Each session lasted for 1.5–2.0 h. The knowledge was assessed before the start of the first session and at the end of the last session using the same questionnaire. The questionnaire comprised 25 and 19 knowledge-based questions with additional information on socio-demographic characteristics for adolescents and YMW, respectively. The knowledge-based questions included questions related to pubertal changes, nutrition, reproductive health, and hygiene of adolescents (Appendix A). Similarly, the questions for YMW included questions on pre-conception care, reproductive health, hygiene, domestic violence, etc. (Appendix A). The questions were asked from the beneficiaries by a team of four investigators. The correct options were given a score of one, and wrong or do not know options were rated zero. The scores of all the 25 or 19 knowledge-based questions (some had multiple options) were summed. Thus, the maximum and minimum scores in the questionnaire for adolescents were 55 and zero, respectively. Similarly, the maximum and minimum scores in the questionnaire for YMW were 26 and zero, respectively. The socio-demographic variables were presented as frequencies or means (±SD) or medians. The mean pre- and post-intervention scores were compared using the paired *t*-test.

The IBM statistical package for social sciences (SPSS) for Windows version 24.0 (IBM Corp., 145 Armonk, New York, NY, USA) was used for data compilation and analysis. For the analysis, the significance level was set at 5%; hence, *p* < 0.05 was considered statistically significant. For the qualitative data obtained from experts on face validity, both inductive and deductive approaches were used in the content analysis. The coded data were collated into themes decided *a priori*. The qualitative data were analyzed manually using Microsoft Excel version 16.0 (Microsoft, Redmond, Washington, DC, USA).

## 3. Results

### 3.1. Development of Flipbook

Considering the need for a comprehensive education material in an interesting and engaging mode, we developed colored flipbooks with illustrations. The desk review and the quantitative data provided us with the key priority areas to be discussed during health education sessions, while keeping in mind the objectives of the project. In total, 76 resource materials from national or international sources were referred during the desk review. After a series of consultations with subject experts and based on evidence from the literature, it was decided to teach all the information on health, nutrition, and hygiene to adolescents or YMW in 12 sessions [16]. The topics of the 12 education sessions for adolescents and YMW are shown in Appendix A. The duration of each session was set at a minimum of 2 h, and the 12 sessions were to be delivered over 12 weeks (one per week). The content of the flipbook was in local (Hindi) language for its ease of use by the peer leaders at the community level. The text was assessed for its use of words, information accuracy, and sentence formation before finalization at three different levels: author, community worker, and language expert. After the finalization of the content, the graphic designer drew illustrations and layout.

The final version of both flipbooks was 18 × 12 inches. Both flipbooks contained 56 double-sided pages with a cover, back cover, and an instruction sheet. In total, both flipbooks had 12 sessions with an opening page for each topic of the session, five key messages, a page for illustration, three to six questions for discussion, and one-liner take away messages on each theme per session (Figure 1 and Figure 2). Moreover, at the end of the session, the one-page story was added to the flipbooks. The stories were based on life skills for adolescents and based on short real-life skills for YMW. The objective of adding the stories was to help participants internalize the learnings from the sessions. The adolescent’s flipbook was yellow in color and titled “*Yauvan*” (adolescence) to symbolize the color of the rising sun (pubertal age). The YMW’s flipbook was green in color and titled “*Sangini*” (partner) to symbolize a woman who is newly married and is the life partner of someone. The flipbook was intended to be used by the ORW and peer leaders while taking sessions in the communities.

### 3.2. Validation of the Flipbook

#### 3.2.1. Face Validity and Concurrent Validity

The comments from the external reviewers and program managers on the face validity were analyzed qualitatively to identify common themes (content-based, illustration-based, and layout-based), and similar responses were grouped. The experts felt that the content was quite appropriate for both flipbooks and needed for the populations, except for a few editions. They suggested highlighting the key points or messages within the sentences based on their relevance. Often in the text, few technical words were requested to be replaced with easier synonyms in both flipbooks. In addition, there were suggestions to change a few stories at the end of the sessions, as they were not related to the theme of the session. One of the reviewers suggested adding the relevant government schemes and elaborating on some of the issues such as domestic violence, post-abortion care, stress management, risk factors for lifestyle diseases, and treatment for human immunodeficiency virus (HIV)/acquired immune deficiency syndrome (AIDS) in the flipbook for YMW. In the illustration sections, most of the reviewers recommended reducing the text. The sequence of a few sessions was changed based on the suggestions. The reviewers suggested that the content in the flipbooks supplemented the existing knowledge or IEC materials available on the government websites.

#### 3.2.2. Content Validity

For the flipbook “*Sangini*”, the mean (SD) age of the ORW was 29 (6) years. There were six males and 36 females. Of the total, 18 (43%) obtained education up to graduation or above, 21 (50%) obtained education up to senior secondary level, and three (7%) ORW were educated up to high school. Eleven out of 12 items were considered adequate by ≥90% participants, as shown in Table 2. The mean score of index I-1 was 0.94. For I-2, 27 participants agreed to adequacy for all 12 items (index = 1), nine participants agreed to 11 out of 12 items (index = 0.91), four agreed to 10 items (index = 0.83), and two participants agreed to eight items (index = 0.66). The mean index score (I-2) was 0.95 (not shown in Table 2).

For the flipbook “*Yauvan*”: the mean index score (I-1) was 0.92. For I-2, 25 participants agreed to adequacy for all 12 items (index = 1), nine participants agreed to 11 out of 12 items (index = 0.91), one agreed to 10 items (index = 0.83), five agreed to nine items (index = 0.75), and one participant agreed to eight items (index = 0.66) and seven items (index = 0.58). The mean index score (I-2) was 0.93 (not shown in Table 2).

#### 3.2.3. Relevance 

Similarly, the relevance index was defined as the proportion of items/chapters agreed to be relevant by each participant.

For the flipbook “*Sangini*”, the index value for all the 15 items was 80% and above, meaning that all 12 sessions and pictures, stories, and key messages at the end of each session were found to be relevant (Appendix A). The mean index score was 0.96.

For the flipbook “*Yauvan*”, the mean index score was 0.95 (Appendix A).

#### 3.2.4. Construct Validity

##### *Yauvan* Flipbook

Adolescent girls: The mean (SD) age of the adolescent girls was 17.5 (1.4) years. Nearly 92% of the beneficiaries belonged to marginalized populations (scheduled castes, SC; scheduled tribes, ST; or other backward classes, OBC), and the remaining 8% belonged to non-marginalized populations. Four out of 50 (8%) girls obtained education below the ninth standard, and the remaining 46 (92%) obtained education up to or above the ninth standard. Thirty-six out of 50 (72%) girls were living in nuclear families, and fourteen (28%) were living in joint families. The median family size was six, and the median monthly family income was INR (Indian rupee) 8000 (USD (United States dollar) $115). As shown in Table 3, the mean (SD) pre- and post-intervention scores were 29.5 (7.8) and 42.0 (5.1), respectively. There was a statistically significant difference in the mean pre- and post-intervention scores (*p* < 0.001). There was no drop-out of the participants during the entire course of the intervention.

Adolescent boys: The mean (SD) age of the adolescent boys was 17.6 (2.7) years. Nearly 92% of the beneficiaries belonged to the marginalized populations (SC, ST, OBC), and the remaining 8% belonged to non-backward classes. Forty-eight out of 50 boys (96%) obtained education beyond the ninth standard, while the remainder (4%) obtained education below the ninth standard. Around half of the boys lived in nuclear families, while the remaining half lived in joint families. The mean (SD) family size was 5.1 (1.4), and the median monthly family income was INR 10,000 (USD 140). The mean (SD) pre- and post-intervention scores were 29.4 (10.4) and 49.8 (3.7), respectively. As shown in Table 3, there was a statistically significant difference in the mean pre- and post-intervention scores (*p* < 0.001). There was no drop-out of the participants during the entire course of the intervention.

##### *Sangini* Flipbook

Young married women: The mean (SD) age of women was 21.5 (2.6). Nearly 96% of the YMW belonged to marginalized populations (SC, ST, OBC), and the remaining 4% belonged to non-marginalized populations. Eleven out of 50 YMW (22%) obtained education up to the ninth standard, while the remainder obtained education beyond the ninth standard (78%, *n* = 39). Ten percent (*n* = 5) of YMW lived in nuclear families, and the remaining 90% lived in joint families. The median family size was six, and the median monthly family income was INR 10,000 (USD 140). The mean (SD) pre- and post-intervention scores were 12.7 (5.5) and 18.7 (2.5), respectively, as shown in Table 4. There was a statistically significant difference in the mean pre- and post-intervention scores (*p* < 0.001). There was no drop-out of the participants during the intervention.

## 4. Discussion

The project “*JAGRITI*” was a multi-site community-based intervention with the components of intensive inter-personal communication, group sessions, mid-media, and awareness events. The prime focus of the project was to enhance knowledge and bridge skill gaps among adolescents and YMW for their improved health, nutrition, and hygiene outcomes. Multiple studies highlighted inadequate knowledge, attitudes, and practices of the adolescents and YMW toward nutrition, reproductive health, and hygiene [20,21,22]. Our study affirms the notion that, despite high literacy levels, there is a lack of adequate knowledge of health education and promotion among the populations [23]. This culminates in poor health and nutrition outcomes with increased morbidities and mortalities [6,24].

Behavior change communication is a proven and sustainable cost-effective approach toward improving health literacy [25]. While delivering intensive health messages as an approach toward BCC, due consideration into their readability, understandability, comprehensibility, and cultural sensitivity needs to be undertaken [26]. The present study aimed to validate the IEC tools for their effectiveness in translating knowledge into action. The flipbooks were assessed as adequate and relevant in all domains by most of the end-users and experts. The participants ranked both the content and the relevance above 80%. The experts were also of the view that flipbooks with such comprehensive content were needed. The flipbooks highlighted some of the unaddressed or inadequately addressed issues in most of the national-level programs such as mental health needs, life skill-based education, and gender-based violence for adolescents, as well as pre-conception care, marital communication, domestic violence, and mental health issues of YMW.

Effective communication is an art that requires appropriate processes, methodology, and materials. There are limited studies that document the piloting process of the education materials and assess their effectiveness after the intervention. One such example comes from a report by Dan Church Aid, Bangladesh. The study evaluated the effectiveness of the IEC materials used to empower migrant workers to improve access to services, including justice [27]. Some of the evidence comes from education aids developed for patient-centered care in hospitals such as leaflets, brochures, etc. [28,29,30]. These validated education materials have the advantage of standardizing information to be shared with the participants and relieving discontent due to contradictory information [31]. Different studies use different methods for validation, such as construct, concurrent, convergent, and predictive validity [32]. However, most of the studies highlighted that the validation process of such materials should be done with the end-users, as well as the beneficiaries of the intervention [31,33,34].

A similar study was conducted to design and validate a module for teaching an integrated approach for yoga therapy to control obesity among adolescents. The validation process included a focus group discussion with 16 subject experts who marked the content validity on a three-point scale. A minimum content validity ratio of 0.5 was considered as the inclusion criteria for the activities in the module [35]. Another study used a mixed-method approach to develop and validate an intervention module for adolescent girls. The intervention combined mental health actions for adolescents with body-focused meditation techniques. In-depth interviews were conducted with six mental health professionals and yoga experts to obtain their preferences for the content and structure of the intervention module. Furthermore, the post-intervention assessment of the module was done using a five-point scale for rating the usefulness of the activities [36]. A similar study from Indonesia used content and empirical validity methods to validate an adolescent peer counselor training module [37].

It can be understood that IEC materials are not the only determinant for improving practices but an indispensable tool in the process. The way in which the education is imparted during counseling sessions and in the appropriate dose and frequency is crucial for the success of the program. The use of validated tools, education materials, and aids in delivering an intervention is an implicit acknowledgement of the effort required to optimize the quality delivery of the intervention [31]. The use of such a validation process assesses the relevance and appropriateness of the education content for greater acceptance and responsiveness by the beneficiaries based on Roger’s diffusion of innovation theory [38]. Moreover, the evidence generated from the feedback by the participants (end-users) enables the integration of learning into local service delivery and especially in complex tailored interventions [39,40,41].

The present endeavor made an effort to point out the relevance of the validation process of tools. In the current scenario, the country is witnessing a multitude of implementation programs by local governmental and non-governmental organizations, and other for-profit agencies. Most of these interventions include health education as an important component. The rigor and intent with which these education tools are validated before reaching millions of people is the need that the programmers have to address.

### Limitations

The study had a few limitations. It was conducted on a small sample size of 42 participants for content validity and relevance, and 50 participants for construct validity. This limits the generalizability of the results. Additional dimensions such as test–retest and inter-rater validity were not performed in the current study, which would have provided evidence for its replication over a period to other implementation areas as well. We limited participation from non-marginalized communities in the assessment, which might have impacted the content in terms of making it more generalizable and sensitive.

## 5. Conclusions

We validated health educational materials (flipbooks) for their content and relevance. The evaluation process of the flipbooks included healthcare professionals, community outreach workers, adolescents, and young married women. The construction of the flipbooks involved scientific knowledge and teamwork, together with designing and layout by artists. The flipbooks are relevant and can be considered as education materials for adolescents and young married women in the communities, in order to bring about changes in their health, nutrition, and hygiene practices. This paper highlights the importance of the validation process as a part of implementation fidelity.

## Figures and Tables

**Figure 1 healthcare-08-00008-f001:**
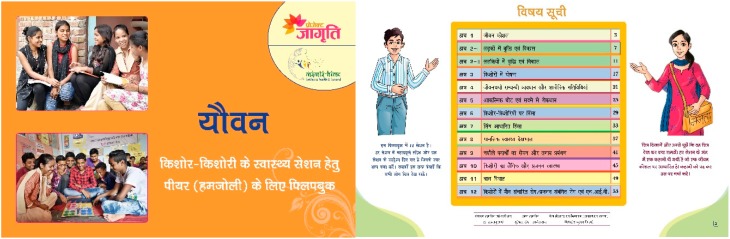
The cover page and the table of contents page in the flipbook for adolescents (“*Yauvan*”). The table of contents displays the name of the 12 structured sessions in the flipbook. The flipbook is in Hindi (local language). The instructions of how to use the book are shown beside the table of contents page.

**Figure 2 healthcare-08-00008-f002:**
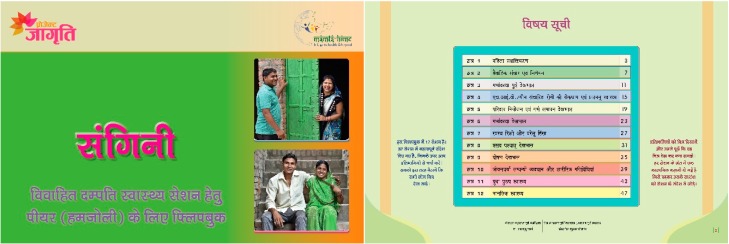
The cover page and the table of contents page in the flipbook for young married women (“*Sangini*”). The table of contents displays the name of the 12 structured sessions in the flipbook. The flipbook is in Hindi (local language). The instructions of how to use the book are shown beside the table of contents page.

**Table 1 healthcare-08-00008-t001:** Parameters for the validation of flipbooks (information, education, and communication (IEC) tool).

S. No.	Parameters of Validation	Purpose	Methodology Sample Population (*n*)
1	Face validity	Does the flipbook appear to achieve what it intends to achieve?	Interview/feedback External reviewers (*n* = 5)
2	Content Validity	Does the flipbook actually explain the topics effectively? Was the tool developed with its content in a user-friendly manner?	12-item-long Likert-based scale Outreach workers (*n* = 42)
3	Construct Validity	Is there an improvement in knowledge after the session was taken with the help of the flipbook?	Pre-post intervention assessment Project beneficiaries (*n* = 50 each for adolescent boys, girls, and young married women)
4	Concurrent Validity	Does the new flipbook add on or supplement the existing tools already given by the agencies/ministries?	Interviews/feedback Experts (*n* = 2)
5	Relevance of the tool	Do the different topics and strategies adopted in the flipbook hold relevance to the needs of the communities?	12-item-long Likert-based tool Outreach workers (*n* = 42)

**Table 2 healthcare-08-00008-t002:** Evaluation of adequacy of agreement on the flipbook content and layout (content validity; I—index). BCC—behavior change communication.

S. No.	Items	*Sangini Flipbook* Adequacy * (*n* = 42) *N* (I **)	*Yauvan Flipbook* Adequacy * (*n* = 42) *N* (I **)
1	The content covered presents relevant information on the health and nutrition needs of the population	42 (1.00)	42 (1.00)
2	Texts seem clear and comprehensive and match the logic, language, and experience of facilitators.	40 (0.95)	39 (0.93)
3	Illustrations or photos or pictures used have a suitable design for understanding.	38 (0.90)	35 (0.83)
4	Illustrations or photos or pictures presented are necessary for understanding the content.	40 (0.95)	36 (0.85)
5	Illustrations and texts motivate the outreach workers to understand the proposed theme.	41 (0.97)	39 (0.93)
6	Applicability of flipbook in BCC sessions	41 (0.97)	40 (0.95)
7	The key messages, in the end, are useful as take-away messages.	41 (0.97)	41 (0.97)
8	The questions in each chapter are easy to understand and help generate inquisitiveness among participants to know the topic.	42 (1.00)	39 (0.93)
9	The stories provided at the end of each session are useful in translating the needs and situations with the application of knowledge.	38 (0.90)	41 (0.97)
10	The two-hour duration of each session and the time allotted for the session are enough and sufficient.	35 (0.83)	37 (0.88)
11	The content is cultural-, gender-, and age-appropriate.	40 (0.95)	36 (0.85)
12	Desired key behavior change points are stressed in the text or emphasized.	40 (0.95)	42 (1.00)
	Mean index score	0.94	0.92

* Number of participants who agreed to the adequacy of each item; ** I: index.

**Table 3 healthcare-08-00008-t003:** Construct validity scores at pre- and post-intervention stage among adolescent girls and boys (*n* = 100).

Knowledge Sections	Adolescent Girls (*n* = 50)			Adolescent Boys (*n* = 50)	
Name of the Sections	Maximum Score *N*	Pre-Test Scores Mean (SD)	Post-Test Scores Mean (SD)	Pre-Test Scores Mean (SD)	Post-Test Scores Mean (SD)
Reproductive health *	24	10.2 (4.1)	17.0 (2.9)	10.6 (4.2)	20.7 (2.2)
Mental health *	6	2.3 (1.7)	4.0 (1.3)	3.0 (1.9)	5.5 (0.8)
Nutrition †	10	7.1 (2.5)	8.0 (1.9)	5.8 (2.9)	9.6 (0.7)
Miscellaneous *	15	9.9 (3.0)	12.9 (1.2)	9.9 (3.6)	13.9 (0.8)
Total score *	55	29.5 (7.8)	42.0 (5.1)	29.3 (10.4)	49.8 (3.7)

Abbreviation: SD: standard deviation; * *p*-value < 0.001 for both boys and girls; † *p*-value < 0.05 for girls and <0.001 for boys.

**Table 4 healthcare-08-00008-t004:** Construct validity scores at pre- and post-intervention stage among young married women (*n* = 50).

Knowledge Sections	Young Married Women (*n* = 50)		
Name of the Sections	Maximum Score *N*	Pre-Test Scores Mean (SD)	Post-Test Scores Mean (SD)
Reproductive health *	11	5.0 (1.9)	6.7 (1.7)
Pre-conception care *	8	4.2 (2.4)	5.8 (1.0)
Miscellaneous *	7	3.3 (2.3)	6.0 (0.9)
Total score *	26	12.7 (5.5)	18.7 (2.5)

Abbreviation: SD: standard deviation; * *p*-value < 0.001.

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
