# Peer review of "Validation of Health Education Material for Youth: A Step to Ensure Implementation Fidelity in Community-Based Interventions"

_healthcare, 2019, doi:10.3390/healthcare8010008_

Round 1
Reviewer 1 Report
This is a generally well written manuscript with a logical structure.
Some reflections are made to the research team trying to improve the quality of the article.
- The tables should be placed according to the rules of the journal.
- The rate of illiteracy in the population should be indicated in the introduction. If this rate is too high, this aspect should be taken into account in the discussion and/or limitations.
The discussion does not look like a discussion as such. Especially the principle, where no ideas related to the results are contributed and no triangulation is made with the results of other research. It would be interesting
Author Response
Point 1: The tables should be placed according to the rules of the journal
Response 1: Yes, I have now places the tables accordingly.
Point 2: The rate of illiteracy in the population should be indicated in the introduction. If this rate is too high, this aspect should be taken into account in the discussion and/or limitations.
Response 2: I have added a line in the introduction “The rate of literacy is around 86% among women (15-24 years) and 93% among men (15-24 years) in the country, which would make health education interventions more successful and sustainable”.
Since the rate of illiteracy is less than 15%, I have not described it in discussion.
Point 3: The discussion does not look like a discussion as such. Especially the principle, where no ideas related to the results are contributed and no triangulation is made with the results of other research. It would be interesting
Response 3: I have now compared in the discussion section the results from our study with the results from other studies. I have added these lines:
A similar study was conducted to design and validate a module for teaching an integrated approach for yoga therapy to control obesity among adolescents. The validation process included a focus group discussion with sixteen subject experts who marked the content validity on a three-point scale. A minimum content validity ratio of 0.5 was considered as the inclusion criteria for the activities in the module [35]. Another study used a mixed-method approach to develop and validate an intervention module for adolescent girls. The intervention combined mental health actions for adolescents with body-focused meditation techniques. In-depth interviews were conducted with six mental health professionals and yoga experts to obtain their preferences for the content and structure of the intervention module. Besides, the post-intervention assessment of the module was done using a five-point scale for rating the usefulness of the activities [36]. A similar study from Indonesia used content and empirical validity methods to validate an adolescent peer counsellor training module [37].
However, I could not add more than this because there is a very limited literature on this, which I have mentioned in the discussion section.

Reviewer 2 Report
The authors are to be commended for their formative and pre-testing research for these two flip books. However the information appears more suitable for presentation as a technical report rather than a journal article - or reduce Tables 1 - 3 and elaborate more on the construct validity test (and note the absence of a control group). Further, there is very little information on how the construct validity was implemented over the 12 sessions ie where were the sessions held, over what time period, what was the drop out rate, etc. Also, was knowledge assessed after each session or at end of 12 sessions?
Author Response
Point 1: reduce Tables 1 – 3
Response 1: I have removed table 3 from the main results and kept it as supplementary Table 4.
Point 2: elaborate more on the construct validity test
Response 2: I have added the analysis of construct validity and added two tables for its description. Also, I have added the supplementary tables 1 and 2 that contain the questions used in the construct validity.
Point 3: note the absence of a control group
Response 3: Yes, I have added a line about this in the methodology section.
“However, there was no control group for comparison”.
Point 4: Further, there is very little information on how the construct validity was implemented over the 12 sessions ie where were the sessions held, over what time period, what was the drop out rate, etc. Also, was knowledge assessed after each session or at end of 12 sessions?
Response 4: I have added lines for this. These are:
“The intervention comprised of 12 education sessions, which were conducted at the youth clubs in the area. The duration of each session was set at a minimum of 2 hours, and the twelve sessions were to be delivered over 12 weeks (one per week). The knowledge was assessed before the start of the first session and at the end of the last session using the same questionnaire. There was no drop-out of the participants during the entire course of the intervention.”

Reviewer 3 Report
Interesting paper but the discussion should be improved underlying the importance of health promotion in different field from parents of newborn to students of primary schools to adolescents or young adults through new technique of communication, multimedia projects or flipbooks or brochures.
1: Vozza I, Capasso F, Marrese E, Polimeni A, Ottolenghi L. Infant and Child Oral Health Risk Status Correlated to Behavioral Habits of Parents or Caregivers: A Survey in Central Italy. J Int Soc Prev Community Dent. 2017 Mar-Apr;7(2):95-99.
2: Vozza I, Fusco F, Corridore D, Ottolenghi L. Awareness of complications and maintenance mode of oral piercing in a group of adolescents and young Italian adults with intraoral piercing. Med Oral Patol Oral Cir Bucal. 2015 Jul
1;20(4):e413-8.
3: Vozza I, Guerra F, Marchionne M, Bove E, Corridore D, Ottolenghi L. A
multimedia oral health promoting project in primary schools in central Italy. Ann Stomatol (Roma). 2014 Nov 20;5(3):87-90.
Author Response
Point 1: but the discussion should be improved underlying the importance of health promotion in different field from parents of newborn to students of primary schools to adolescents or young adults through new technique of communication, multimedia projects or flipbooks or brochures.
1: Vozza I, Capasso F, Marrese E, Polimeni A, Ottolenghi L. Infant and Child Oral Health Risk Status Correlated to Behavioral Habits of Parents or Caregivers: A Survey in Central Italy. J Int Soc Prev Community Dent. 2017 Mar-Apr;7(2):95-99.
2: Vozza I, Fusco F, Corridore D, Ottolenghi L. Awareness of complications and maintenance mode of oral piercing in a group of adolescents and young Italian adults with intraoral piercing. Med Oral Patol Oral Cir Bucal. 2015 Jul
1;20(4):e413-8.
3: Vozza I, Guerra F, Marchionne M, Bove E, Corridore D, Ottolenghi L. A
multimedia oral health promoting project in primary schools in central Italy. Ann Stomatol (Roma). 2014 Nov 20;5(3):87-90.
Response 1: I have added a para in the discussion section. Also, I have added one of these studies in the discussion section. Since, the other two studies were not very close to my study so could not include them.
“Our study affirms the notion that despite high literacy levels, there is a lack of adequate knowledge of health education and promotion among the populations [23]”
Vozza, I.; Capasso, F.; Marrese, E.; Polimeni, A.; Ottolenghi, L. Infant and child oral health risk status correlated to behavioral habits of parents or caregivers: a survey in central Italy. J. Int. Soc. Prev. Community Dent. 2017, 7, 95–99.
“A similar study was conducted to design and validate a module for teaching an integrated approach for yoga therapy to control obesity among adolescents. The validation process included a focus group discussion with sixteen subject experts who marked the content validity on a three-point scale. A minimum content validity ratio of 0.5 was considered as the inclusion criteria for the activities in the module [35]. Another study used a mixed-method approach to develop and validate an intervention module for adolescent girls. The intervention combined mental health actions for adolescents with body-focused meditation techniques. In-depth interviews were conducted with six mental health professionals and yoga experts to obtain their preferences for the content and structure of the intervention module. Besides, the post-intervention assessment of the module was done using a five-point scale for rating the usefulness of the activities [36]. A similar study from Indonesia used content and empirical validity methods to validate an adolescent peer counsellor training module [37].”

Round 2
Reviewer 2 Report
The authors have done a very good job in developing and pre-testing their materials - but that should be done with all education materials -- and is more appropriately presented as a technical report.Author Response
Comments: The authors have done a very good job in developing and pre-testing their materials - but that should be done with all education materials -- and is more appropriately presented as a technical report.
Reply: Thanks for your comments. I wanted to publish it as a peer-reviewed article and not just as a technical report. Peer-reviewed articles are valued more or cited quite often because they had undergone a scientific assessment compared to technical reports.
